# The Triumph of the Spin Chemistry of Fullerene C_60_ in the Light of Its Free Radical Copolymerization with Vinyl Monomers

**DOI:** 10.3390/ijms25021317

**Published:** 2024-01-21

**Authors:** Elena F. Sheka

**Affiliations:** Institute of Physical Researches and Technology, Peoples’ Friendship University of Russia (RUDN University), 117198 Moscow, Russia; sheka@icp.ac.ru

**Keywords:** virtual polymerization, spin theory of fullerenes, free radical polymerization, digital twins, energy graphs, thermodynamic and kinetic descriptors, polymerization passports, vinyl monomers, stable radicals, fullerene C_60_

## Abstract

The spin theory of fullerenes is taken as a basis concept to virtually exhibit a peculiar role of C_60_ fullerene in the free radical polymerization of vinyl monomers. Virtual reaction solutions are filled with the initial ingredients (monomers, free radicals, and C_60_ fullerene) as well as with the final products of a set of elementary reactions, which occurred in the course of the polymerization. The above objects, converted to the rank of digital twins, are considered simultaneously under the same conditions and at the same level of the theory. In terms of the polymerization passports of the reaction solutions, a complete virtual picture of the processes considered is presented.

## 1. What Is It, the C_60_ Fullerene Molecule?

Forty years ago, fullerene C_60_ burst into the science of molecules, marking an unexpected reward for all natural scientists for their many years of honest, persistent, and selfless work. Mathematicians were delighted with the amazing gracefulness of the spatial structure of the molecule [1,2]; chemists could not get enough of the unique variety of its chemical properties [3,4,5]; physicists sang Hallelujah to its magnetism [5,6,7]; biologists admired its biocompatibility with living organisms [8,9,10]; physicians enthusiastically engaged in research into new drugs [5,10,11]; geologists have been stubbornly looking for it in natural dumps of amorphous carbon [12]. Having come as an extraterrestrial from the Cosmos [13,14], the molecule happily basked in the atmosphere of love and universal worship.

All this taken together formed the basis of the question put forward as a heading of the introductory section. The general answer is simple and obvious: it is unique. It is unique, existing in reality, once presented by Fuller’s truncated icosahedron C_60_ digital twin virtually [13]. It is empirically unique because it is composed of atoms of a unique element, the only one allowed by Nature to form covalent bonds of three different types; because it represents a nanosize molecule of exactly known structure and chemical composition. It is unique theoretically, because its electron theory involves spin, drastically differing from standard spinless molecular science. It is unique in being simultaneously an empirical and virtual spin molecule. Regioselective chemistry, physical magnetism, free-radical-like medicine, and biology contributed to strengthening the belief in the spin character of this empirical molecule. The electronic theory, represented by its Quantum Chemistry, found itself under severe sanctions due to the difficulties of taking spin into account in an many-electron system (see a profound discussion of the problem in [15]). Scientists were faced with a choice—either to abandon the *ab initio* configuration interaction (CI) approach, replacing it with a simplified one, but allowing the consideration of many-electron systems taking spin into account, or to continue to adhere to first principles, but abandon the virtualization of the C_60_ fullerene. A compromise was proposed, to limit the first-principles configurational interaction of electrons with spins to the main CI contributor—the unrestricted Hartree–Fock (UHF) two-determinant approximation [16,17,18,19,20,21,22]—and to replace the procedure for the exact calculation of many-electron integrals by determining parameterized contributions to this interaction, calculated in a multipole approximation [23,24]. In this case, the spin component of the electronic system was taken into account with maximum accuracy. The creators of this semi-empirical approach, Prof. Dewar and his school, did a great deal of work showing that the error of the semi-empirical method does not account for more than a small percentage, using examples of admissible simultaneous consideration of electronic systems ab initio and semi-empirically. As expected, the first consideration of the C_60_ molecule in the semi-empirical UHF approximation turned out to be successful [5,25,26,27,28,29]. The radical nature of the molecule was established virtually, opening up the possibility of creating a self-consistent spin theory of the molecule.

An obstacle to the transformation of this starting into the dominant approach to virtual fullerenics arose from an unexpected direction. Its root was the active development of another approximate method for calculating many-electron systems, based on electron density. The concept of density functionals, well-developed by this time, provided an important advantage over semi-empirical HF methods, allowing one to work with a wider atomic composition of molecules [30]. Gradually, the DFT method became the main method for the virtual quantum chemistry of large molecular systems. Encouraged by its success, users easily transferred it to both fullerenes and graphene molecules, without thinking about electron spins. At the same time, as it turned out, both the DFT and UDFT methods do not sense electron spins well and are practically spin-free techniques [31,32]. To date, there is not a single virtual DFT examination of the molecule radicality that either confirms or predicts the reality, while the number of studies of, generally speaking, open-shell molecules does not decrease. Evidently, there has been no DFT spin theory of fullerene C_60_ until now.

In contrast, the UHF spin theory of fullerenes has evolved over the years into a coherent scientific concept of radical molecules in general. The main objects of its application are a large family of nanocarbons, covering fullerenes, graphene molecules, and carbon nanotubes [33,34,35]. The theory made it possible to explain and predict almost all the known spin characteristics of these materials [36]. The past years have expanded the wide range of phenomena over topochemistry and reaction kinetics [37]. In all the cases, the spin theory not only made it possible to explain many problematic issues, but also offer reliable predictions, which were then confirmed experimentally.

Although a great deal of radical chemistry has been carried out, the theory has so far left aside such an important chemical process as free radical polymerization of organic monomers [38]. These amazing polymers, victors and winners in the fierce intermolecular competition of numerous radical products, turned out to be very sensitive to the presence of C_60_ fullerene in the relevant reaction solutions (RSs) [39,40,41,42,43,44,45,46,47,48,49,50,51,52,53,54,55,56,57,58]. The feature irrefutably evidences the radical nature of this fullerene, on the one hand, and gives rise to the temptation to test to what extent the spin theory of fullerene is capable of providing a self-consistent description of the numerous nuances of this process, on the other. At the same time, virtual copolymerization of vinyl monomers in the presence of C_60_ has not been practically considered until recently. A relatively small number of individual DFT calculations [59,60,61,62,63,64,65,66,67,68,69] have not solved any of the existing issues, demonstrating their complete helplessness. The situation that arose turned out to be challenging for the spin theory of fullerene molecules, aimed at proving its deep conceptuality and self-consistency. In this paper, it will be shown that the spin theory not only perfectly senses and responds to the slightest changes occurring in RSs, but also makes it possible to predict the results of behaviors of such media that have not yet been studied. Our consideration is based on the Digital Twins (DTs) concept [70], which allows one to play a virtual game with numerous event participants [71], providing the detection of trends characteristic of these radical events.

## 2. A Short Sketch of the UHF Spin Theory of Fullerenes

### 2.1. The Root of the C_60_ Fullerene Radicality

Digital twin Buckminsterfullerene C_60_ of $C_{i}(I_{h})$ continuous symmetry [72] represents a closed carcass structure of covalent *sp*^2^C-C bonds, each of which is uniquely dynamically unstable. The bond remains stable until its length is less than a critical value $R_{crit}^{C=C}$ of 1.395 Å (see Figure 1) [35,73]. Above this range, it turns from an ordinary covalent bond uniting two atoms into a covalent bond between gradually radicalized atoms. The bond elongation is accompanied by the breaking of spin symmetry because the α and β spins are located on different electron orbitals [18,19,20,21,22] and react on the bond’s elongation differently. This feature generates a non-zero spin density matrix, the real and imaginary parts of which present the spin density and spin current density, responsible for spin emergents [15], among which the total, $N_{D}$, and partial in relation to each atom, $N_{DA}$, number of effectively unpaired electrons [21,22] take a particular place [36]. Both emergents present quantitative measures of the bond radicalization, while thereby determining the molecular chemical susceptibility (MCS) of the molecule as a whole and atomic chemical susceptibility (ACS) of individual atoms, respectively. For singlet all-carbon molecules with an even number of electrons, the numbers of electrons with different spins are $N_{\alpha }=N_{\beta }$ and $N_{D}=∆{\hat{\left(S\right)}}^{2}$, where the quantity in brackets is the squared spin. Both $N_{D}$ and $N_{DA}$ are amenable to computational and experimental verification performed many times [36]. They are both quantitative indicators of both non-zero spin density availability and dynamic instability of the *sp*^2^C-C bonds.

Figure 2 summarizes what was said above and exhibits a significant dispersion of both long and short bonds of the molecule (Figure 2a) as well as bond-length compositions of both groups (Figure 2b) when bond radicalization is taken into account. As expected, the presence of a limited number of bond-length groups is tightly connected with a peculiar distribution of spin density over the molecule atoms, the value of which in terms of $N_{DA}$ is presented in Figure 2c, while its multi-color image is shown in the insert. As seen in Figure 2c, the C_60_ DT is regioselective towards intermolecular interaction in entering the field of chemical attacks. Evidently, the stronger the interaction, the higher is $N_{DA}$, so that the ACS is the best descriptor to be put into the ground of the algorithm of the virtual derivatization of the fullerene. The figure also exhibits a deep conceptual difference between the DTs of the C_60_ fullerene from the viewpoint of spin-symmetrical RHF and DFT approaches, in contrast to the spin-unsymmetrical UHF one.

### 2.2. Spin-Density Algorithm of the Virtual Derivatization

The first application of the spin theory to one of the basic events of the molecule chemistry—to the molecule derivatization—reveals the next uniqueness of the C_60,_ which concerns the unavoidable *sp*^2^→*sp*^3^ transformation, related to the targeted atom of the molecule core. To mark derivatives below we will use the term fullerenyls, suggested earlier [74,75] and subsequently supported [61,62,69], mono- and polyfullerenyls to be more exact. The bond transformation affects not only all covalent bonds of this atom, but other closely surrounding atoms. Resisting the imposed deformation of the structure and minimizing its influence, which is aimed at preventing the opening of the core structure, the entire network of covalent bonds comes into motion, relaxing in a new equilibrium position of all its atoms. Figure 3 exhibits an example of such events. Presented in the form of a two-dimensional plot, shown in the figure, it does not fully reflect the structural change in the valence bonds of the molecule, but concerns only their structural-passport part. The C_60_ DT atom number 33 is subject to attack by the free radical ${AIBN}^{•}$, which is the most often expected event in the free radical polymerization (FRP) of vinyl monomers when C_60_ is added to the RS (see [57] and references therein). The action leads to the formation of monofullerenyl FR, shown in the figure. As a result, the number of valence bonds of atom 33 increases to four, and their type changes from *sp*^2^ to *sp*^3^, because of which the length of the entire bond increases. However, only the latter, highlighted in the figure, is a bond of this type that connects a fullerene with a radical. As for the other three bonds, they remain in the system of *sp*^2^ bonds of other atoms, thus becoming of mixed type. This peculiarity is manifested in the fact that on the terminal atoms of the bonds (atoms 24, 32, and 34) the spin density does not vanish, as on atom 33, but takes on different values, which is expressed in the ACS $N_{DA}$ values of 0.20, 0.36, and 0.55 *e* in fullerenyl instead of 0.20, 0.27, and 0.27 *e* in the original fullerene. Each atom of the considered trio is the terminal one of two more bonds, different in each case, so that the $N_{DA}$ values represent the total effect of a complex structural–spin density rearrangement of the molecule, caused by the addition of one addend through a single *sp*^3^C-C bond. Thus, each act of molecule derivatization causes a strong collective reflection, which explains the exclusive lability of the molecule structural, electronic, and spin systems that is one more uniqueness of the C_60_ fullerene.

Lilac balls of different size on the FR image mark the first five atoms from the top of the ACS Z→A list of the $N_{DA}$ values of the fullerenyl, from 0.55 to 0.29 *e.* These atoms form a set of targets for the next addition, being located at a one-, two-, three-, and four-bond distance from atom *A.* Which of these atoms participates in the next step of the derivatization depends on the addend structure, because of the mandatory requirement to avoid sterical hindrances when coupling with the fullerene core. Each addend is of individual case, thus completing the ACS top-list algorithmic choice with a bond-number distance from the place of the previous anchoring. Thus, in the case of the second coupling of ${AIBN}^{•}$ radical to the fullerenyl FR, the free-hindrance addition occurs on the fifth target atom, located at a four-bond distance from atom A. The remaining four atoms are ‘chemically sleeping’ in this case. The third ${AIBN}^{•}$ location on the C_60_ core will occur on the ACS top-list atom at a four-bond distance from the second anchoring and the further derivatization will be continued similarly. Evidently, the maximal number of ${AIBN}^{•}$ radicals coupled to the C_60_ core is 12.

### 2.3. Donor-Acceptor Peculiarity and Dry Polymerization of Fullerene C_60_

One more uniqueness of the C_60_ molecule concerns its donor and/or acceptor (DA) abilities, both of which are strong [5,76]. This feature greatly influences the intermolecular interaction that accompanies molecule polymerization [77], controlling the height of barriers on the standard energy graphs that are characteristic for any chemical reaction [71]. For the first time, this was manifested in the course of the C_60_ fullerene virtual dimerization [76], thus including reaction kinetics in the area of responsibility of the spin theory of the molecule.

The word group “C_60_ fullerene and polymerization” appeared almost simultaneously with the presentation of the molecule as a particularly attractive object of molecular physics and chemistry. From these distant times, we were talking about two different routes of study, the first of which concerned the dry polymerization of the all-carbon C_60_ monomers, accompanied by the formation of 1D-, 2D-, and 3D-configured polymer chains ${\left(C_{60}\right)}_{n}$. Started in 1995 with C_60_ photopolymerization [78] and continued by the application of other physical techniques, such as high temperature and pressure [79,80], electric voltage [81,82], electron beams [83], and so forth, the first route led to the foundation of an extended study of both real and virtual magnetism of solid fullerites. The main scientific and technological problems of the latter were solved by 2006 (see [6,7,84,85,86]). The second route was traditionally that of liquid chemistry, and concerned studies of wet polymerization, when either introducing fullerene into already known RSs, ensuring the polymerization of already known monomers, or generating new fullerene-containing monomers and stimulating the polymerization of the latter. In contrast to the first case, wet polymerization does not concern all-carbon C_60_, but mainly its fullerenyls. A brief but intelligent overview of these processes can be found in [87].

Over the thirty-year history of C_60_ fullerene polymerization, the spin theory of molecules was used twice to explain the features of this process: firstly, in the case of a dry all-carbon homogeneous oligomerization of C_60_ fullerene from a dimer to tetramers [76,77], and, secondly, in the case of wet FRP of vinyl monomers [88,89,90,91]. In the first case, a computational technique was used for the first time to determine not only the coupling energy $E_{cpl}$ of the final reaction product, but also the activation energy $E_{ad}$ of its decomposition. This technique concerns the construction of the decomposition barrier profile that visualizes a standard energy graph of each elementary reaction when presenting the total energy of the intermolecular complex E(R) as the function of the reaction coordinate. The first application of the technique, related to the C_60_ dimerization, is presented in Figure 4a.

The spin theory of fullerenes suggests a definite scheme of the expected successive oligomerization of C_60_ molecules when going, say, from dimer to tetramer within the ${\left(C_{60}\right)}_{n}={\left(C_{60}\right)}_{n-1}+C_{60}$ oligomerization scheme, as shown in Figure 4b. According to the ACS top-list $N_{DA}$ of dimer ${\left(C_{60}\right)}_{2}$, there are four pairs of top $N_{DA}$ atoms, which are marked by red balls in the lower right corner of the figure. The first two pairs combine the most reactive atoms adjacent to the cycloaddition (contact-adjacent or *ca* atoms). The next four atoms are located in the equatorial planes of both monomers (equatorial or *eq* atoms). In spite of the high chemical reactivity of the former atoms, they are not accessible in the course of further oligomerization, so that *eq* atoms of both monomers are actual targets. Following these $N_{DA}$ indications, a right-angle triangle trimer (90^0^-trimer) must be produced. Therefore, not the ‘pearl necklace’ configuration, intuitively suggested as the most expected for C_60_ oligomerization [87], but a more complicated 2D one is favorable for trimerization.

Similarly, the high-rank $N_{DA}$ atoms of the trimer, as seen in Figure 4b, form an incomplete *ca* pair of the highest activity and three pairs of *eq* atoms of comparable activity. Three tetramer compositions that follow from this ACS indication related to the trimer are shown in the figure. None of them belongs to the ‘pearl necklace’ family, thus presenting 2D tetramers 1 and 2 and 3D tetramer 3. Among the latter, tetramer 1 possesses the highest $E_{cpl}$ and is expected to continue the oligomerization, offering its high-rank $N_{DA}$ atoms, marked by red balls, as targets for the next C_60_ addition. Those form six pairs of the most active *ca* atoms and four pairs of *eq* atoms, the position of which dictates the continuation of oligomerization as the formation of 3D configurations of pentamers. Therefore, the formation of large plane membranes of polymerized C_60_, which the latest DFT experiments are devoted to [92,93], does not seem likely.

As seen in the figure, the C_60_ dimerization is a barrier reaction with a quite high barrier ($E_{a}$ = 23.11 kcal/mol), which explains why the reaction does not occur spontaneously and requires rather rigid measures, such as photoexcitation, high temperature, and high pressure, as well as application of an electric field to overcome the barrier and provide the dimerization [76]. At the same time, it was shown that the barrier height is mainly determined by $E_{gap}=I_{A}-\varepsilon_{B}$. Passing to oligomers ${\left(C_{60}\right)}_{n}$, one faces a peculiar situation, characteristic for fullerenes. As it turns out, both the ionization potential $I_{A}$ and electronic affinity $\varepsilon_{B}$ of the ${\left(C_{60}\right)}_{n}$ oligomer only slightly depend on *n* and practically coincide with those related to the monomer molecule. This has been computationally justified for oligomers of complex structure, characterized by *n* varying up to 10 [85]. Consequently, the graph related to $C_{60}+C_{60}$ dyads determines a general behavior of both ${\left(C_{60}\right)}_{n-1}+C_{60}$ and ${\left(C_{60}\right)}_{m}+{\left(C_{60}\right)}_{k}$ dyads at each successive step of oligomerization.

The second application to the spin theory of the C_60_ molecule polymerization has occurred just recently [88,89,90,91] concerning its wet polymerization, which we now move on to discuss.

## 3. Wet Polymerization of Fullerene C_60_

### 3.1. A Short Sketch of Empirical Observations

This concerns rigorous studies of the kinetics of the initial stage of the FRP of vinyl monomers in the presence of the C_60_ fullerene, sometimes mentioned as free radical copolymerization (FRCP) of vinyl monomers with the fullerene. A large amount of empirical data related to the FRP of vinyl monomers as well as to their FRCP with *TEMPO* and C_60_ fullerene [39,40,41,42,43,44,45,46,47,48,49,50,51,52,53,54,55,56,57,58] make it possible to offer a systematic view of this chemical process, presented in several graphs in Figure 5.

The figure accumulates empirical data related to the time-dependent percentage conversion x(t) of a monomer that well represents the kinetics of the initial stage of the FRP of vinyl monomers and their FRCP with fullerene C_60_ and *TEMPO*, while being initiated with either ${AIBN}^{•}$ or ${BP}^{•}$ free radicals [56,57,58]. The panorama presents a complex of experiments performed under the same conditions concerning the temperature, solvent, monomer content, and chemical content of free and stable radicals. Graphs labeled 1 in all the panels present the referent FRP of the relevant monomer. Figure 5a–c exhibit the effect of small additives of C_60_ on the referent FRP of methyl methacrylate (MMA), styrene (St), and *N*-isopropyl acrylamide (NIPA). As seen, the effect is different for the monomers, once showing itself as decreasing the slope of the referent graph 1 for MMA and appearing as a delay in the start of FRP of styrene and NIPA due to the appearance of low-intensity fractions on their graphs. Until recently, these fractions, observed in other cases as well, were designated in a single way, referring them to the induction periods. Figure 5d,e show the effect of the combined action of C_60_ and *TEMPO* on the FRP of MMA and St. As seen in the figures, the presence of *TEMPO* manifests itself in the same way in both cases, providing the appearance of a fairly long induction period. But the effect of small additions of C_60_ is still different, completely similar to that observed in the absence of *TEMPO* (Figure 5d,e), but delayed in relation to the beginning of the relevant monomer FRP by the duration of the induction period. Analysis of the data presented in Figure 5a,b,d,e shows that the stable radicals *TEMPO* and C_60_ act superpositionally on the polarizable medium. Figure 5f demonstrates that the manner in which C_60_ affects the FRP of a monomer depends not only on the monomer but also on the type of free radical involved in the reaction. Thus, in the case of replacing ${AIBN}^{•}$ with ${BP}^{•}$, FRCP of St with C_60_ is accompanied by a significant restructuring of the low-intensity fraction of the complete graph $x\left(t\right),$ pointing to a change in the kinetics of the process. The discussed effects are well-pronounced and strong, stimulating a persistent desire to understand their subtleties from the standpoint of the spin theory of radicals.

### 3.2. Virtual FRCP of Vinyl Monomers with C_60_ Fullerene in Light of the Spin Theory of the Molecule

The first summarized view on the virtualization of FRCP of vinyl monomers with C_60_ fullerene has been presented recently [91]. The approach is based on a number of fundamental concepts, among which there are the following. The pilot concept concerns the presentation of polymerization process as a chain reaction [94,95] involving a set of elementary reactions. The latter are considered as independent and superpositional, thereby allowing the use of all the accumulated experience in the quantum chemical consideration of reactions [96,97,98,99,100,101]. Both initial reagents and final products of the reactions form the DTs pool that is the main object of both the theoretical and virtual consideration. Two types of descriptors are introduced, thermodynamic and kinetic ones, which are equally characteristic of all elementary reactions. The descriptors’ unification makes it possible to issue polymerization passports, which are a ‘personal identifying document’ for each virtual reaction solution (VRS) and provide a potential comparison of all the virtual characteristic features of the polymerization events under study with empirical reality. It would be logical to introduce a VRS as a source of the relevant DTs in each case under study. A large amount of empirical data related to the FRP of vinyl monomers [38,102] as well to their FRCP with *TEMPO* and C_60_ fullerene [39,40,41,42,43,44,45,46,47,48,49,50,51,52,53,54,55,56,57,58,103] greatly facilitates the VRS design.

*List of the elementary reactions.* A rather complete list of the relevant elementary reactions related to the FRCP under study is presented in Table 1. Nominations listed in the table concern simultaneously both reactions and their final products. The first common characteristic of the reactions is their radical character. However, they are therewith distinctly divided into two groups that cover bimolecular combination reactions (1) and (2), uniting free radicals with monomers, and grafting bimolecular reactions (3)–(12), that in the case of the stable-radical C_60_ are reactions of the fullerenyl design. Products of the first group of reactions, as well as those of reactions (3b) and (4), are free radicals, while those of the second group are either stable species or fullerenyl stable radicals in the case of *TEMPO* and C_60_, respectively.

Reactions (1) and (2), uniting a free radical with a monomer in monomer-radical $RM^{•}$ and its oligomers $RM_{n}^{•}$, evidently govern the FRP of monomers. The former is the cornerstone of the entire polymerization process, determining its feasibility as such. Reactions (3a) and (3b) open the list of actions connected with the presence of the C_60_ fullerene in each studied VRS. This reaction doublet reflects one more of the unique properties of the fullerene concerning the intermolecular junctions formed by two *sp*^2^C-C bonds, one belonging to fullerene, while the other presents a vinyl group of monomers. Accordingly, the junction can be either two-dentant or one-dentant. If the first configuration causes the formation of a [2 × 2] cycloadded monoadduct stable radical FM, similar to the patterned C_60_, the second results in the formation of a fullerene-grafted monomer radical $FM^{•}$, similar to $RM^{•}$. Equilibrated structures of the two DTs associated with methyl methacrylate in Figure 6 are supplemented with the ACS distributions, revealing the radical properties of the bodies expressed in terms of the unpaired-electron fractures $N_{DA}$. The picture is common for all the representatives of vinyls. ACS plottings of both DTs are settled over the background, presenting the ACS distribution related to the initial fullerene C_60_. As seen in the figure, both fullerenyls retain the *multi*-target type of the radicalization. The appearance of new targets with increased radicality in the fullerene core (see the most prominent atoms 35, 22, and 11 in Figure 6a, and extra atoms 37 and 35 in Figure 6b) is the expected consequence of the reconstruction of the fullerene *sp*^2^C-C bond system, caused by the fullerenyl formation discussed earlier with respect to Figure 3. As previously, lilac balls present the three top-list ACS values in both cases. Bright lilac balls mark targets of the next adduct, located at a three-bond distance from the atoms of the first anchoring. The emergence of a new target ability of fullerenyl $FM^{•}$ with a predominant $N_{DA}$ of 0.97 *e* at atom 62, related to the vinyl bond of the monomer (see Figure 6b), exhibits the undeniable readiness of the latter to continue the association with other monomer molecules, similar to what is observed in the case of a standard FRP [71]. Accordingly, reaction (2) describes the polymer chain growth $RM_{n}^{•}$ initiated with a free radical, while reaction (4) ${FM}_{n}^{•}$ describes the monomer polymerization, once grafted on fullerene. Although the existence of a reaction (3b) was suspected in a number of cases, a confident conclusion was not made, and this reaction as well as reaction (4) were classified as unlikely. The first proof of the fullerene-initiating polymerization of styrene has been obtained just recently [89,91], and will be considered in Section 4.

Reaction SM (5) reveals the capturing of a monomer with a one-target stable radical. In contrast to the above fullerenyls, SM species, when it is formed, presents a routine non-radical one-bond-coupled intermolecular complex. In contrast to a non-reactive monomer, the capturing of its monomer-radical ${RM}^{•}$, described by reactions FRM (6) and SRM (7), is traditionally highly expected in both cases. Actually, these reactions are of particular importance, having the opportunity to completely stop the polymerization process. Then, reactions FR (8) and $SR\;\left(9\right)$ follow, revealing a similar capturing of free radicals $R^{•}$. Both reactions evidently affect the monomer polymerization, decreasing the number of initiating free radicals. Reaction SR (10) takes into account the interaction of stable radicals between themselves, while reactions RFM (11) and SFM (12) describe the capturing of monomer-radical $FM^{•}$ with stable ones. This set of elementary reactions is quite complete for the consideration of the initial stage of both FRP of vinyl monomers and their FRCP with stable radicals. The relevant DTs of their final products alongside with input ingredients form a large pool.

*Polymerization of fullerenyls.* The reaction list in Table 1 does not include the polymerization of either C_60_ itself or its fullerenyls. As for the former, earlier it was said that its ‘dry’ polymerization is highly difficult because of the great height of the corresponding barrier in Figure 4a. The corresponding activation energy $E_{a}$ constitutes 23.11 kcal/mol. However, in a ‘wet’ medium of RSs the situation may change, particularly with respect to various fullerenyls filling a working reactor. To quickly answer this question, it is appropriate to recall the unique role of the DA contribution to the intermolecular interaction between C_60_ fullerene monomers. The energy gap $E_{gap}=I_{A}-\varepsilon_{B}$ quantifies this contribution, and has been shown to influence the height of the barrier. To verify this conclusion, Table 2 shows the calculated $E_{gap}$ values of two sets of fullerenyl oligomers. The first set corresponds to fullerenyls of the FR type, which presents a coupling of free radical $AIBN^{•}$ with the C_60_ fullerene and belongs to the organic content being the final product of reaction (8). The second fullerenyl belongs to the metalloorganic content, and presents the monoadduct composed of the C_60_ coupled with butyl lithium [104,105,106,107]. Figure 7 shows the equilibrium structures of the DTs corresponding to monomers, dimers, and tetramers of these fullerenyls. In both cases, the DTs design was controlled by the ACS algorithm, which masterfully revealed atom targets for the sequenced steps of the monomers’ polymerization.

Evidently, the linear oligomer compositions of both fullerenyls drastically changed from the square-nest one of the (C_60_)_n_ shown in Figure 4b. Oligomers of both monomers are composed as alternate chains of one-dentant and two-dentant intermolecular junctions. Evidently, still more important nuances of the fullerenyl oligomerization will be discovered, both virtually and experimentally. Nevertheless, two important commonalities of the molecules must be noted immediately. As seen in Table 2, $E_{gap}$ in both oligomer families is kept within not more than 1.5% accuracy, practically constant and equal to that of monomers. Its value for organic and metalloorganic species differs by about 1 eV, thus causing the lowering of the barrier height for metalloorganic fullerenyls by about 22 kcal/mol, which makes the oligomerization of the latter practically barrier-free. The difference retains unchanged for all other organic fullerenyls studied in the project.

The experimentally discovered dramatic increase in the rate of polymerization of (BuLi)n-C_60_ [104,105,106,107] convincingly indicates a decrease in the barrier. A similar phenomenon was previously observed when comparing the polymerization of C_60_ and AC_60_ alkali fullerites [108]. It was suggested in [76] that the effect is caused by the presence of alkaline metals with low $I_{A},$ which crucially decreases $E_{gap}$ of the AC_60_ species. As for organic fullerenyls, their $E_{gap}$ turned out to deviate from the reference C_60_ value only slightly, thus pointing to fullerenyl polymerization as kinetically unfavorable, and thereby removing the issue of including the oligomerization of fullerenyls in the elementary reactions presented in Table 1 from the agenda.

*Molecular descriptors of free-radical polymerization.* Molecular descriptors, introduced for a facilitation of the simultaneous consideration of as many elementary reactions from Table 1 as possible, led to the foundation of the next concept of the polymerization digitalization [91]. Following the energy graph, describing each elementary reaction and, in particular, presented in Figure 4a, two quantities $E_{cpl}$ and $E_{ac}\equiv E_{a}$ were suggested as thermodynamic and kinetic descriptors of the virtual FRP of vinyl monomers. The latter concerns the standard description of the rate constant,$\;k\left(T\right),$ which is expressed through the Arrhenius relation [96,97,98,99,100,101] as:(1)$$k\left(T\right)=Ae^{\left(\frac{-E_{a}}{kT}\right)}.$$

Here, A is a complex frequency factor, while $E_{a}$ presents the activation energy of the bimolecular combination reaction, $E_{ac}$. The main difficulty in the constant evaluation is provided by the highly complicated nature of frequency factor A. Its determination concerns basic problems of the rotational-vibrational dynamics of polyatomic molecules, such as the great number of both vibrational and rotational degrees of freedom as well as their anharmonicity. However, for one-type elementary reactions, A is expected to change weakly [96,97,98,99,100,101], so that the activation energy becomes governing and is suggested as the kinetic descriptors for FRP and/or FRCP of vinyl monomers [88,89,90,91].

*Virtual polymerization passports.* The next concept, greatly facilitating the polymerization digitalization, concerns the polymerization passports (PPs) which are issued to every VRS [91]. Similarly to other identifying personal documents, PPs consist of two pages. The first of them contains textual information concerning the VRS, which involves the nomination of elementary reactions and the corresponding DTs, supplemented with thermodynamic and kinetic descriptors, $E_{cpl}$ and $E_{a}$, respectively. The second page reveals photo-images of the considered DTs equilibrated structures. This PP form turned out to be quite suitable for a comparative study of the digitalized predictions as well as for their verification with available empirical data.

Table 3 accumulates identical fragments of the PPs’ text pages related to different VRSs. The elementary reaction content is limited to the FRP of the monomers, as well as to their FRCP with the C_60_ fullerene. Only kinetic descriptors are involved. Figure 8 presents a fragmentary image page of the passports related to the elementary reactions occurring in the considered VRSs in the presence of the fullerene. The complete content of the issued PPs can be found elsewhere [88,89,90,91]. When configuring both the table and the figure, the data follow the ordering suggested by the empirical data presentation in Figure 5.

The collection of DTs shown in Figure 8 is composed in the following way. Species of the first row present monomers M and monomer-radicals ${RM}^{•}$, grouped in pairs for methyl methacrylate (MMA), styrene (St), and N-isopropyl acrylamide (NIPA), respectively. The three columns thus formed include pairs of FM and ${FM}^{•}$ monofullerenyls (the second row) and single FRM monoaggucts (the third row). All the DTs are related to elementary reactions where the alkyl-nitrile ${AIBN}^{•}$ acts as a free radical. The fourth row includes three DTs related to the reactions ${RM}^{•}$, FRM, and FR that are related to the VRS of styrene with the benzoyl peroxide ${BP}^{•}$ as a free radical. A complete list of the DTs considered during the project can be found in [91].

## 4. Main Results Related to the Initial Stage of the Free-Radical Copolymerization of Vinyl Monomers with C_60_ Fullerene

Analyzing the data presented in Table 3 and Figure 8, the following conclusions can be made.

1. The set of elementary reactions involving the fullerene C_60_ is common for all the vinyl VRSs, and consists of four members that are reactions FM, ${FM}^{•}$, FRM (${FR}^{A}M$ and ${FR}^{P}M$, once initiated with either ${AIBN}^{•}$ or ${BP}^{•}$ free radicals, respectively), and $FR\;({FR}^{A}$ and ${FR}^{P}$, as in the previous case). The considered fullerenyls FM, ${FM}^{•}$, FRM, and FR are monoadducts, designed according to the spin theory of C_60_ when anchoring the relevant addends at the same core atom in all cases. The general nature of the structure of fullerenyls and the dominant role of fullerene in them allows us to speak about the similar type of the associated reactions, which makes it possible to use the kinetic descriptors listed in Table 3 to quantify the rates of the corresponding reactions.

2. According to the kinetic descriptors of the MMA VRS, the fullerene-associated reactions form a series of the following order concerning their rate constants:(2)$$k_{R}^{F}>k_{rm}^{F}\gg k_{1m}^{F},\;k_{2m}^{F}.$$

The reaction of the capture of the free radical by fullerene heads this series, and has every reason to be carried out first. It does not compete with the FRP of MMA, and occurs in parallel with the latter, while reducing the current concentration of free radicals in the VRS, which should affect the reduction in the rate of monomer conversion during its polymerization. It is this response of the graph x(t) to the presence of fullerene in the RS that is observed experimentally (see Figure 5a). Obviously, the decrease in the conversion rate depends on the fullerene concentration and increases with its growth, which, as seen in the figure, really takes place experimentally.

3. The series that orders the sequence of the fullerene-associated reactions in the VRS of styrene drastically differs as per the discussion above. It is important that the latter depends greatly on the initiating free radical. Thus, in the case of the ${AIBN}^{•}$ being in service (the VRS^A^ case), the series looks like the following:(3)$$k_{1m}^{F}\geq k_{R}^{F}>k_{rm}^{F},$$
giving a clear advantage to the formation of the monomer radical ${FM}^{•}$ initiated by the fullerene. Actually, this reaction competes with reaction (1), and the final result of the competition depends on the ratio between rate constants $k_{i}$ and $k_{1m}^{F}$. The ratio of the corresponding kinetic descriptors listed in Table 3 shows the comparability of both rates. However, additional reasoning presented in [91] tends to suggest that:(4)$$k_{1m}^{F}\geq k_{i},$$
so that the polymerization in the VRS^A^ is started with the styrene polymerization linked to and occurring at the fullerene body. Until the reaction proceeds, no other polymerization that involves styrene will start. When all the fullerene content is resumed, a standard FRP of styrene, initiated with free radical ${AIBN}^{•}$, will occur. Therefore, the total conversion graph x(t) should consist of two sections related to fullerene- and ${AIBN}^{•}$- stimulated FRP of styrene. Since under usual conditions the [C_60_] content is lower than the [St] one by three orders of magnitude, the initial section of the x(t) graph should be very low in amplitude, which is actually observed in reality (see Figure 5b). Evidently, the time-duration of this fracture of the graph increases when the [C_60_] grows, which is fully evident in Figure 5b. Noteworthy is the preservation of the slope of the main linear part of the graph, which remains the same as in the reference reactor without the addition of fullerene.

As seen in Table 3, the substitution of ${AIBN}^{•}$ with ${BP}^{•}$ changes the situation drastically. The series (3) in the case of VRS^P^ takes the form:(5)$$k_{rm}^{F}>k_{1m}^{F}\gg k_{R}^{F},$$
and the first place goes to the reaction (6) of the absorption of the ${RM}^{•}$ monomer-radical by fullerene. Naturally, this reaction has a direct impact on the polymerization of styrene, intervening between the two main reactions (1) and (2) of its polymerization. According to Table 3 and the previous discussion [91], the sequence of reactions follows the ordering:(6)$$k_{i}>k_{rm}^{F}>k_{p},$$
so that the formed monomer-radical ${RM}^{•}$ is captured with the fullerene, which prevents the FRP of the monomer propagation, thus providing the presence of a zero-amplitude conversion graph x(t) and revealing a classical induction period until the [C_60_] content is resumed. The FRP of styrene starting at the point is fully identical to that of the reference VRS^P^. It is this type of conversion graph that is presented in Figure 5f.

4. The fullerene-associated sequence of reaction of the VRS of NIPA follows the series:(7)$$k_{rm}^{F}\gg k_{R}^{F}\gg k_{1m}^{F},\;k_{2m}^{F}.$$

Reaction (6), once barrier-free, absolutely dominates, so that each of newly formed monomer-radical ${RM}^{•}$ is doomed to immediate capture. The long induction period, which determines the complete resuming of [C_60_], becomes the hallmark of the conversion graph x(t) of the VRS, which is observed experimentally (see Figure 5c).

5. As seen in Table 3, the variety of the fullerene-associated reactions provides a reasonable choice of the fastest. The latter are characterized by kinetic descriptors whose values do not exceed 9–10 kcal/mol. It is quite evident that the reactions with higher descriptors remain outside the polymerization process. This is another important evidence of the absence of polymerization of fullerene and its fullerenyls under the practical reaction conditions described in the caption to Figure 5. The kinetic descriptor for the polymerization of organofullerenes exceeds 20 kcal/mol, which makes it practically impossible as well.

6. As shown in [91], in contrast to a large variety of the participation of the C_60_ fullerene in the FRCP of vinyl monomers, the one-target stable radical *TEMPO* behaves much more modestly. This variety of the above-discussed VRSs is characterized by the only reaction concerning the body. This is reaction (7), which describes the capture of all the studied monomer-radicals ${RM}^{•}$ with *TEMPO*. As shown above, this reaction is characterized by a clearly seen zero-amplitude conversion graph x(t), attributed to the induction period. Predictions obtained virtually find their full confirmation in experiments, which is shown in Figure 5d,e. Since the radicals *TEMPO* and C_60_ do not interact, their action in experimental RSs is fully superpositional, which is clearly seen in the figures.

## 5. Star-Branched Polymers of the Fullerene C_60_ Generated in the Course of Its FRCP with Vinyls

The discussions presented in the previous sections have concerned the initial stage of the chemical transformation occurring in a reaction solution. The picturesque varying phenomena caused by the presence of fullerene C_60_ are not limited only to them, even after the words we have said about the “complete resumption of the [C_60_] content by the end of reactions (4), (6) and (8)”. All three fundamental reactions of the initial period are terminated in the general case by the formation, not of the relevant monofullerenyls $FM_{n}^{•}$, FR, and FRM, but of m-polyadducts $F{\left(M_{n}^{•}\right)}_{m}$, $FR_{m}$, and $F{\left(RM\right)}_{m}$. The radical activity of fullerene polyadducts ${F\left(X\right)}_{m}$ gradually decreases when m grows. This does not happen immediately but through many successive steps, so that in the case of the simplest addends, which are atomic hydrogen and fluorine, ${F\left(X\right)}_{m}$ adducts cease to be radicals when m takes the value 36 in the case of hydrogen [109] and 48 in the case of fluorine [110]. Naturally, when the addends are of complex structure, deradicalization is generated faster, since the number of anchoring seats for such addends on the carbon core of the molecule is strictly limited by the condition of preventing steric hindrances.

Analysis of numerous experimental data concerning the FRCP of vinyl monomers with C_60_ fullerene shows that the radical activity of fullerenyls in the initial period is not completely extinguished, so that the latter take part in the subsequent formation of the polymer mass of the monomer under study. One commonly hears that the fullerene is incorporated into the polymer chains of the final product, becoming a star-branching center. From the viewpoint of the spin theory of fullerenes, the assumption seems to be quite reasonable. In fact, while remaining radicals, fullerenyls ${F\left(X\right)}_{m}$ actively interact with other radicals available in the RS, the main part of which during the period of the massive polymerization is presented with oligomers of the basic monomer $RM_{k}^{•}$, thus producing complex compositions of the ${F\left(X\right)}_{m}{\left(RM_{k}^{•}\right)}_{m\prime }$ type. The variety of compositions is too large for a reliable prediction to be made. Evidently, any progress in this direction can be expected only after general algorithms of the fullerene C_60_ polyderivatization, complimented with thorough kinetic analysis, are developed, which, in turn, requires in-depth study of both experimental and theoretical concepts. The main problem concerns the determination of the numbers m and m′, as well as revealing the location of target atoms of the fullerene core supply anchoring addends X and $RM_{k}^{•}$ to the core. To demonstrate the difficulties to be met on this path, Figure 9 exhibits the formation of the three simplest possible stars, based on the new knowledge of the VRSs discussed in the previous sections when retaining the individuality of the FRCP processes in all cases.

The C_60_-star ${F\left(R^{A}\right)}_{3}{\left(R^{A}MMA_{5}\right)}_{1}$ in Figure 9a accumulates two main reactions, the capture of three initiating radicals ${AIBN}^{•}$ in the initial stage of the FRCP of methyl methacrylate with fullerene C_60_, and the addition to the fullerenyl core of the oligomer-radical $R^{A}MMA_{5}^{•}$. The DT design is subordinated to control by the ACS algorithm. As seen in the figure, the oligomer-radical is characterized by being the only target whose reactivity is provided with $N_{DA}$ of 0.98 *e*. The first fullerene-based reaction of the MMA VRS concerns the free radical trapping. According to Figure 3, the trapping is followed by the reconstruction of both *sp*^2^C-C bond set and spin density on their atoms. Pale lilac balls of different sizes mark the top five atoms from the fullerenyl ${F\left(R^{A}\right)}_{1}$ Z→A list of the $N_{DA}$ values from 0.55 to 0.29 *e.* The first four of these atoms form a set of targets for the next addition, but being located at a one-, two-, and three-bond distance from atom *A* are not accessed for the radical $R^{A}$ addition because of steric hindrance, thus forming a four-atom set of ‘chemically sleeping’ atoms, And only the fifth bright lilac atom at a four-bond distance from atom A is ready to adopt the addition of the second radical $R^{A}$.

This addition completes the previous four-atom set of sleeping atoms of ${F\left(R^{A}\right)}_{1}$ with one more four-atom set of sleeping atoms of ${F\left(R^{A}\right)}_{2}$ of the $N_{DA}$ values from 0.47 to 0.30 *e*, providing the opportunity only for the ninth bright lilac atom, located at a four-bond distance from atom B, to accept the third radical $R^{A}$. The sleeping atom set of ${F\left(R^{A}\right)}_{3}$ is enlarged up to 12 of the $N_{DA}$ values from 0.55 to 0.28 *e*, and the 13th atom of $N_{DA}$ = 0.27, positioned four bonds away from atom C, is ready to service a target atom for the next addition. It is this atom to which the oligomer radical $R^{A}MMA_{5}^{•}$ is added, thus providing the formation of the first branch of the MMA C_60_-star ${F\left(R^{A}\right)}_{3}{\left(R^{A}MMA_{5}\right)}_{1}$.

In the case of the NIPA C_60_-star ${F\left(R^{A}M\right)}_{1}{\left(R^{A}NIPA_{8}\right)}_{1}$ in Figure 9b, the main reaction of the initial period concerns the trapping of monomer-radical ${R^{A}M}^{•}$. Similarly to the previous case, the action is followed by the formation of the four-atom sleeping set of the $N_{DA}$ values from 0.55 to 0.29 *e*, while the fifth atom of $N_{DA}$ = 0.29 *e* is the target of the next addition of the oligomer radical, that is $R^{A}NIPA_{8}^{•}$ in this case. A similar analysis allows us to detect the target atom of the NIPA C_60_-star ${F\left(R^{A}M\right)}_{1}{\left(R^{A}NIPA_{8}\right)}_{1}$ that is ready for the next addition of either ${R^{A}M}^{•}$ or $R^{A}NIPA_{8}^{•}$.

The formation of the styrene C_60_-star ${F\left({St}_{6}\right)}_{1}{\left(R^{A}St_{6}\right)}_{1}$, shown in Figure 9c, proceeds similarly to the above two cases. In spite of its composition, ${St}_{6}$ is much more cumbersome than $R^{A•}$ and $R^{A}M^{•}$, and the reaction of spin density of the fullerenyl ${F\left({St}_{6}\right)}_{1}$ on the addition is practically identical to the previous cases. Again, four of the five carbon atoms with the highest $N_{DA}$ are sleeping, while the fifth one accepts the addition of the $R^{A}{St}_{6}^{•}$ oligomer-radical. The latter, in its turn, detects new five target atoms.

However, this simple four-bond algorithm will not always take place. Evidently, it will retain in the case of successive additions of radical $R^{A•}$, thus forming the polyadduct ${F\left(R^{A}\right)}_{n}$ of the biggest n = 12. Apparently, this can be expected for the polyadduct ${F\left(R^{A}M\right)}_{n}$. In the case of large cumbersome oligomer-radicals, the algorithm will be violated because of a considerable worsening of the spatial configurations of the added addends, which prevents avoidance of steric hindrances. Thus, the number of added branches will not exceed 4–6, depending on the concrete spatial configuration of the addends.

## 6. Concluding Comments

As befits a serious, responsible radical, C_60_ fullerene immediately becomes a participant in any radical reaction. Having unique properties, the molecule brings to each case a special charm that is unique to it. From this point of view, free radical polymerization of vinyl monomers demonstrates these molecules’ capabilities in the best possible way. Experimental fullerene polymerics is diverse and surprising. Virtual molecular polymerics provides a broad springboard for testing the basic concepts underlying fullerene physico-chemistry. This article presents the results of a spin theory as applied to the elementary reactions that form the backbone of the polymer process. The main objects of the theory are digital twins. As expected, the main application of the theory concerns the algorithmic design of DTs, the success of which determines the conclusions drawn and predictions made. As a result, it turned out to be possible to comprehensively explain the entire set of features of polymerization in the initial stage that are stimulated by fullerene presence. Thermodynamic and kinetic descriptors were determined that make it possible to assess the degree of competition of various elementary reactions involved in this process. The main directions of development of the spin theory of molecules, aimed at implementing the virtualization of the final processes of formation of a polymer product, were identified as well. To date, this area of chemical science has turned out to be the most ready for digitalization, which can be considered the main achievement of the spin theory of radicals, in general, and fullerene, in particular, the results of practical work on which laid the foundations of the current article.

## Figures and Tables

**Figure 1 ijms-25-01317-f001:**
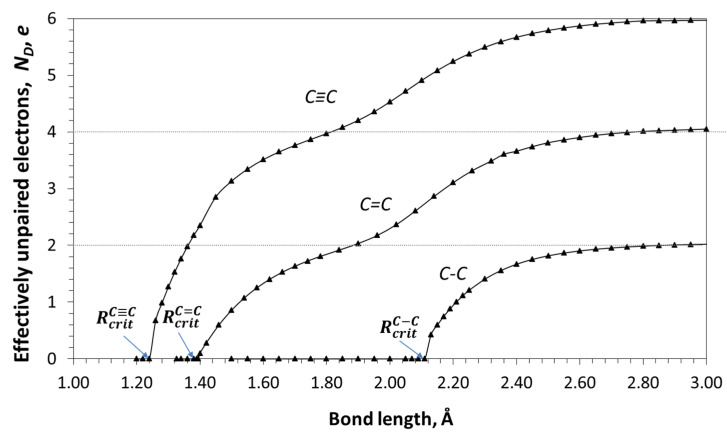
Dynamic instability of covalent bonds in ethane (*sp*^3^C-C bond), ethylene (*sp*^2^C-C bond), and propyne (*sp*^1^C-C bond) molecules. UHF AM1 calculations. Digitalized data are taken from Refs. [35,62].

**Figure 2 ijms-25-01317-f002:**
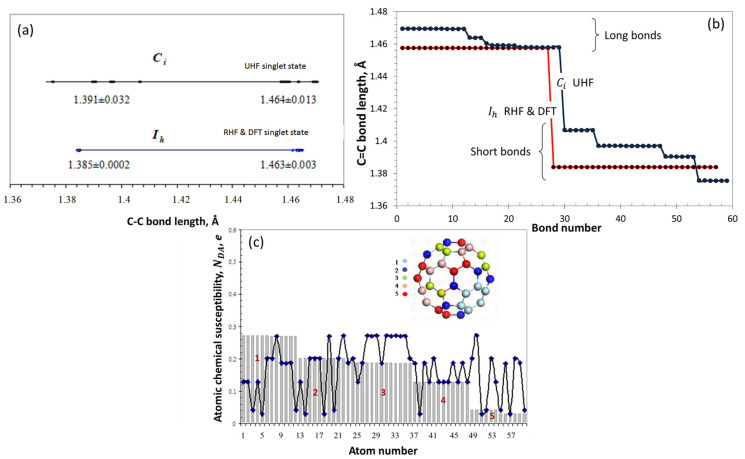
Dynamic instability of the *sp*^2^C-C bonds (**a**,**b**) and atomic chemical susceptibility (**c**) of the C_60_ fullerene. Digits of colored balls in insert and on histogram are the same.

**Figure 3 ijms-25-01317-f003:**
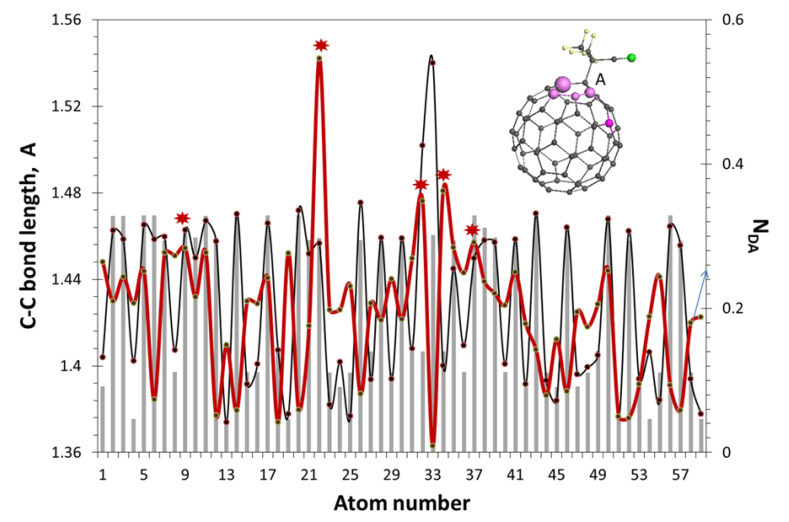
The distribution of the covalent C-C bond length in C_60_ fullerene (histogram). The same (dark-gray curve with dots) and that of the ACS (red curve) in fullerenyl FR are shown in the insert. Red asterisks mark atoms visualized with lilac marking in the insert. The structural-passport set of bonds is the same for both C_60_ and FR. Small yellow and gray balls mark hydrogen and carbon atoms, respectively. Larger green ball depicts nitrogen atom. UHF AM1 calculations.

**Figure 4 ijms-25-01317-f004:**
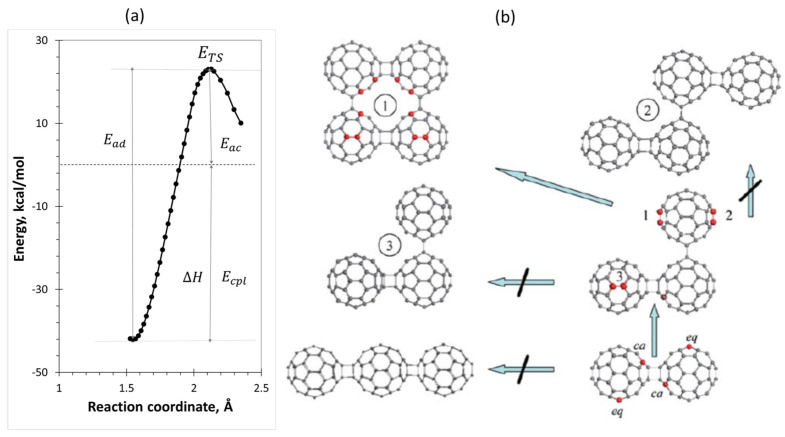
(**a**) Barrier profile of the dimer (C_60_)_2_ decomposition. (**b**) Stepwise oligomerization of C_60_ from dimer to tetramer. Equilibrium structures. Crossed arrows indicate unfavorable continuations. Red balls mark target atoms of the highest $N_{DA}$ values. Coupling energies constitute −42.23 kcal/mol (dimer); −74.73 kcal/mol (trimer); −164.63 kcal/mol (tetramer 1); −13.84 kcal/mol (tetramer 2); −117.66 kcal/mol (tetramer 3). Digitalized data of Ref. [77]. UHF AM1 calculations.

**Figure 5 ijms-25-01317-f005:**
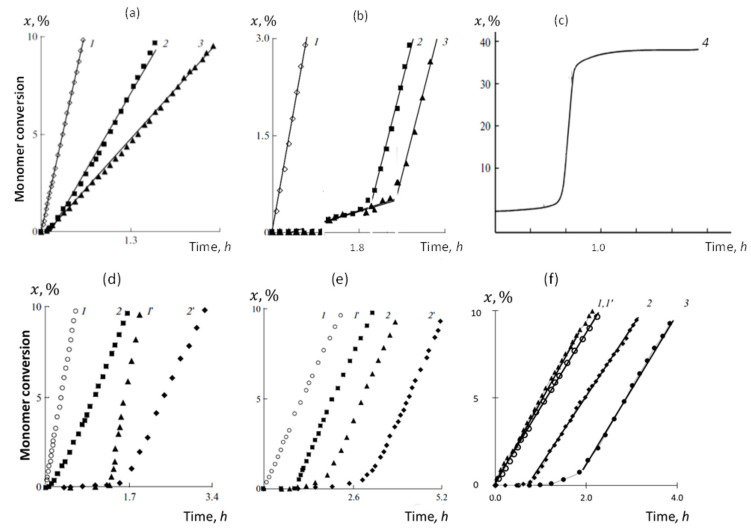
Empirical kinetics of the initial stage of both FRP of vinyl monomers and their FRCP with *TEMPO* and fullerene C_60_. $AIBN^{•}$-initiated conversion of methyl methacrylate (**a**), styrene (**b**), and *N-*isopropyl acrylamide (NIPA) (**c**) in the presence of different [C_60_]: 0 (graphs labeled 1); 1.0 × 10^–3^ (graphs labeled 2); 2.0 × 10^–3^ mol/L (graphs labeled 3); 6.7 × 10^–3^ mol/L (graphs labeled 4). The same for methyl methacrylate (**d**) and styrene (**e**), but with [*TEMPO* (C_60_)] 0 (graph labeled 1); [*TEMPO*] 1.0 × 10^–3^ mol/L and [C_60_] 0 (graph labeled 1′); [*TEMPO*] 0 and [C_60_] 1.0 × 10^–3^ mol/L (graph labeled 2); [*TEMPO* (C_60_)] 1.0 × 10^–3^ mol/L (graph labeled 2′). (**f**) Conversion of styrene, initiated with $AIBN^{•}$ (graphs labeled 1′ and 3) and $BP^{•}$(graphs labeled 1 and 2) in the absence (1′ and 1) and in the presence (3 and 2) of fullerene [C_60_] = 2.0 mol/L. T = 60 °C; o-DCB solvent; [MMA(St)] = 2.0 mol/L; [NIPA] = 0.73 mol/L; [*AIBN* (*BP*)] = 2.0 × 10^−2^ mol/L. Digitalized data of Refs. [56,57,58].

**Figure 6 ijms-25-01317-f006:**
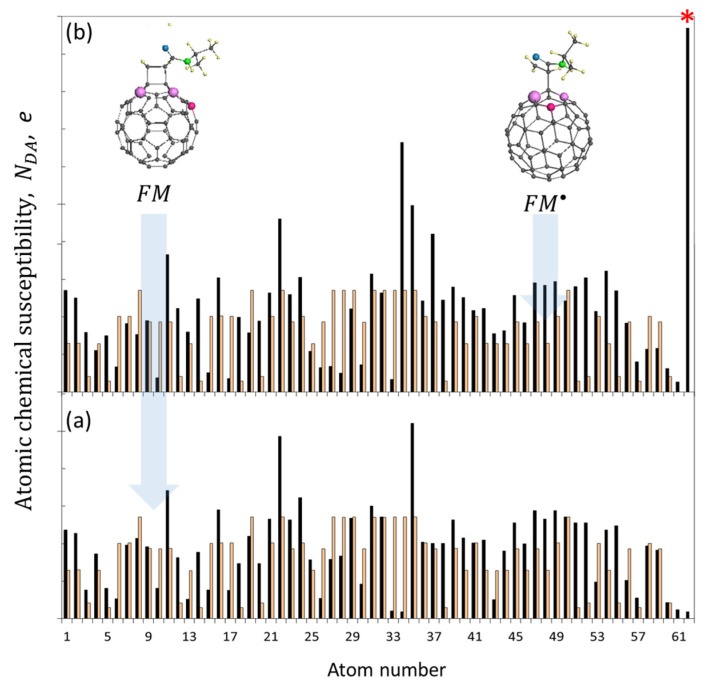
Equilibrium structures and ACS radicality of methyl-methacrylate fullerenyls FM (**a**) and ${FM}^{•}$ (**b**) (black histograms). Light rose marks the ACS radicality of fullerene C_60_. The carbon-atom numeration of fullerene and fullerenyls is the same. Small yellow and gray balls mark hydrogen and carbon atoms, respectively. Larger green and blue balls depict nitrogen and oxygen atoms. Red asterisk marks the target atom of the vinyl group of the fullerenyl ${FM}^{•}$. UHF AM1 calculations.

**Figure 7 ijms-25-01317-f007:**
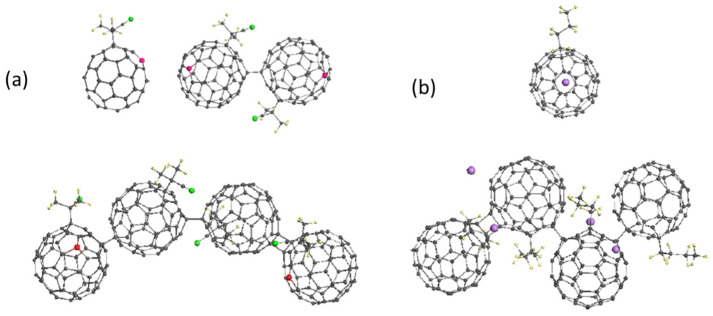
Equilibrium DTs related to the oligomerization of organic (**a**) and metalloorganic (**b**) fullerenyls. Red balls mark targets of the next additions. UHF AM1 (**a**) and PM3 (**b**) calculations.

**Figure 8 ijms-25-01317-f008:**
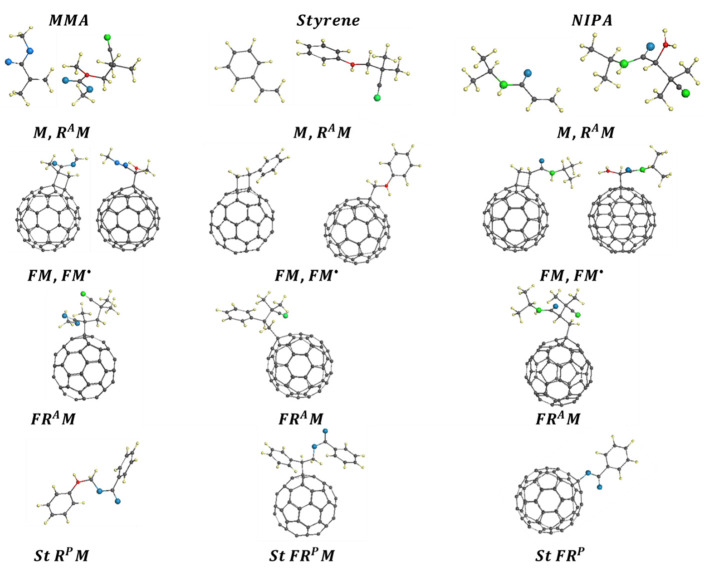
Equilibrium structures of digital twins related to the FRCP of methyl methacrylate (MMA), styrene (St), and *N*-isopropyl acrylamide (NIPA) with C_60_ fullerene. The DT nomination follows that of Table 1. Small yellow and gray balls mark hydrogen and carbon atoms, respectively. Larger green and blue balls depict nitrogen and oxygen atoms. Red balls mark carbon target atoms. UHF AM1 calculations.

**Figure 9 ijms-25-01317-f009:**
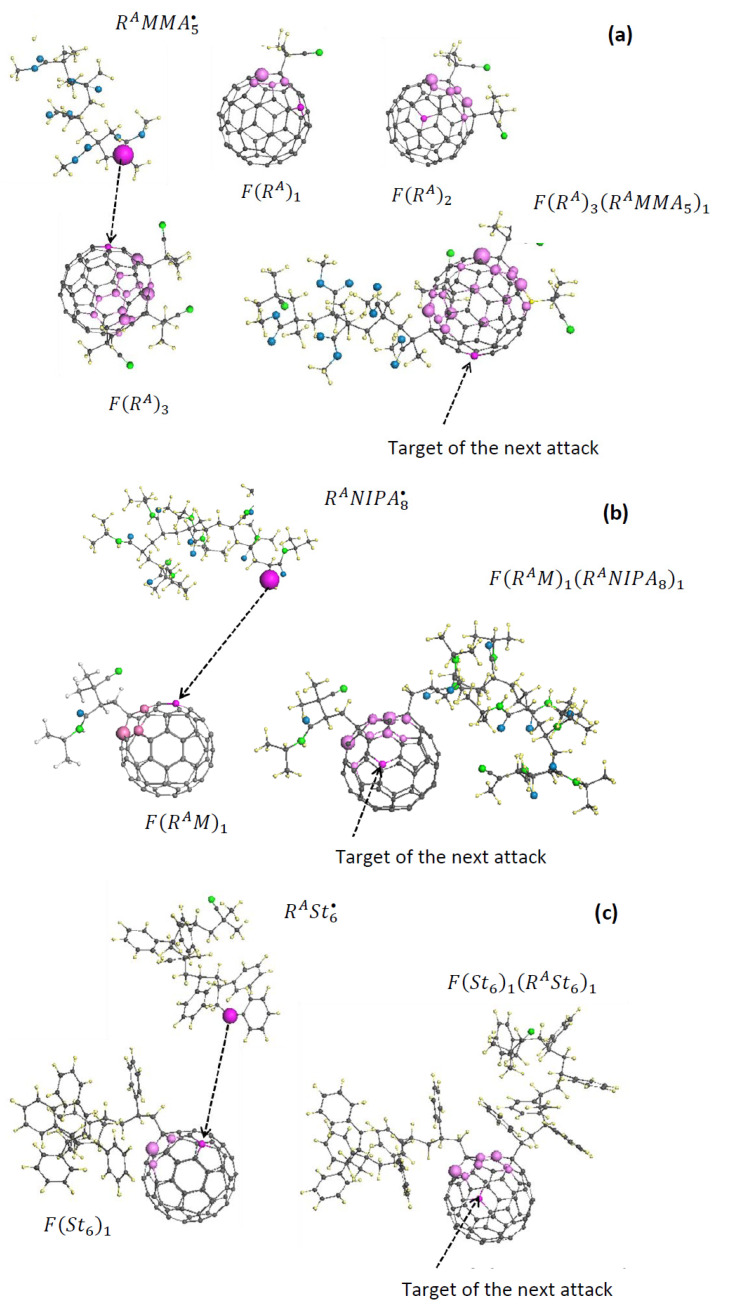
First steps of the C_60_-branched star polymer formation in VRSs of methyl methacrylate (**a**), NIPA (**b**), and styrene (**c**). Sizes of lilac balls correspond to the ACS values on the atoms. UHF AM1 calculations.

**Table 1 ijms-25-01317-t001:** Nomination of elementary reactions and/or digital twins related to the initial stage of the free-radical copolymerization of vinyl monomers with stable radicals.

Reaction Mark	Reaction Equation ^(1)^	Reaction Rate Constant	Reaction Type
(1)	$R^{•}+M\to RM^{•}$	$k_{i}$	generation of monomer-radicals
(2)	$RM^{•}+(n-1)M\to RM_{n}^{•}$	$k_{p}$	generation of oligomer-radicals, polymer chain growth
(3a)	$F^{•}+M\to FM$	$k_{2m}^{F}$	two-dentant grafting of monomer on C_60_
(3b)	$F^{•}+M\to FM^{•}$	$k_{1m}^{F}$	one-dentant stable radical grafting of monomer, generation of monomer-radical
(4)	$FM^{•}+(n-1)M\to FM_{n}^{•}$	$k_{p}^{F}$	generation of oligomer-radical anchored to C_60_, polymer chain growth
(5)	$S^{•}+M\to SM^{•}\equiv SM$	$k_{1m}^{S}$	one-dentant coupling with monomer
(6)	$F^{•}+RM^{•}\to \;FRM$	$k_{rm}^{F}$	monomer-radical grafting on C_60_
(7)	$S^{•}+RM^{•}\to \;SRM$	$k_{rm}^{S}$	monomer-radical capturing with stable radical
(8)	${F^{•}+R}^{•}\to FR$	$k_{R}^{F}$	free radical grafting on C_60_
(9)	${S^{•}+R}^{•}\to SR$	$k_{R}^{F}$	free radical capturing with stable radical
(10)	$F^{•}+S^{•}\to FS$	$k_{S}^{F}$	stable radical grafting on C_60_
(11)	$R^{•}+FM^{•}\to RFM$	$k_{FM}^{R}$	monomer-radical $FM^{•}$ capturing with free radical
(12)	$S^{•}+FM^{•}\to SFM$	$k_{FM}^{S}$	monomer-radical $FM^{•}$ capturing with radical S

^(1)^ M, R,F, S mark vinyl monomers, initiating free radicals (either ${AIBN}^{•}$ or ${BP}^{•}$, see detailed description in [71,92]), or stable radicals (fullerene C_60_, and *TEMPO*), respectively. Superscript black spot distinguishes radical participants of the relevant reactions.

**Table 2 ijms-25-01317-t002:** Energy gap $E_{gap}=I_{A}-\varepsilon_{B}$, eV, characteristic for oligomers of the fullerene C_60_ derivatives ^(1)^.

Oligomers	Fullerenyl FR	C_60_ + Butyl Li
Monomer	−7.07	−5.96
Dimer	−7.13	−5.52
Trimer	−6.99	−5.40
Tetramer	−6.88	−5.41

^(1)^ Referent $E_{gap}$ values of monomer C_60_ are −7.22 eV and −7.02 eV in the AM1 and PM3 UHF versions of the CLUSTER-Z1 program, respectively.

**Table 3 ijms-25-01317-t003:** Elementary reactions and DTs of their final products, supplemented with virtual kinetic descriptors related to the FRCP of vinyl monomers with the C_60_ fullerene, kcal/mol ^(1)^.

	M		$R^{A}M^{•}$	${AIBN}^{•}$ ($R^{A•}$)	$C_{60}$ (F)	
Digital Twins’ set						
M	$M_{n}$		${\;R}^{A}M_{n}^{\cdot •\cdot }$	$R^{A}M^{•}$	Two-*dentant*	One-*dentant*
					FM	$FM^{•}$
$R^{A}M^{•}$	-		-	-	${FR}^{A}M\;$	
$R^{A•}$	-		-	-	${FR}^{A}$	
$FM^{•}$	$FM_{n}^{\cdot •\cdot }$		-		-	
Virtual reaction solution of methyl methacrylate (MMA)						
M			**10.46** (2) ^(2)^	**12.28** (1) ^(2)^	>20 (2) ^(3)^	$E_{ad}\ll E_{ac}$
$R^{A}M^{•}$			-	-	**11.58**	
$R^{A•}$			-	-	**9.40**	
Virtual reaction solution of styrene (St)						
M			**12.06** (2) ^(2)^ 6.12–16.49 (2–6) ^(2,4)^	8.50 (1) ^(2)^	**24.52** (2) ^(3)^	**8.38** (1)
$R^{A}M^{•}$	-		-	-	**9.73**	
$R^{P}M^{•}$					**7.02** ^(5)^	
$R^{A•}$	-		-	-	**9.41**	
$R^{P•}$	-		**2.79** (1) ^(2,5)^ **8.78** (2)	-	**28.82** ^(5)^	
$FM^{•}$	**11.25** (2) ^2)^		-		-	
Virtual reaction solution of *N*-isopropyl acrylamide (NIPA)						
M			**8.39** (2) ^(2)^ 7.74 (3) ^(2,4)^ 8.49 (4) ^(2,4)^	**19.09** (1) ^2)^	**17.29** (1) ^(3)^ **27.79** (2)	**20.01**
$R^{A}M^{•}$			-	-	**0.023**	
$R^{A•}$			-	-	**9.398**	

^(1)^ Bold data are determined from the decomposition barrier profiles [91]. ^(2)^ Digits in brackets mark the number of monomers in the oligomer chain. ^(3)^ Digits in brackets indicate one-bond and two-bond decomposition of FM fullerenyl (see details in [91]). ^(4)^ The data are calculated by using the Evans–Polanyi–Semenov relation presented in [71]. ^(5)^ The data are related to the VRS of styrene when free radical ${AIBN}^{•}$ is substituted with ${BP}^{•}$.

## Data Availability

Any data or material that support the findings of this study can be made available by the corresponding author upon request.

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
