# Peer review of "The Triumph of the Spin Chemistry of Fullerene C60 in the Light of Its Free Radical Copolymerization with Vinyl Monomers"

_ijms, 2024, doi:10.3390/ijms25021317_

Round 1
Reviewer 1 Report
Comments and Suggestions for Authors
This is a very personal and inspiring review of a scientific mission to understand the overall chemistry, the spin properties, the free radical polymerizations and other elementary reactions of the fullerene C60 nano-size molecule. The complexity of the molecule makes the decision to choose the appropriate theoretical quantum mechanical method a delicate matter. It is adequately demonstrated that the unrestricted Hartree-Fock theory, through its simplicity, yet have the capability to capture the intrinsic nature of the fullerene radicality. This incorporates bond transformation effects, donor-acceptor peculiarities, wet polymerizations, etc., as well as in other important processes, where an exposed sensitivity to the presence of C60 in the relevant reactions are expressed. The main results display several interesting and important conclusions followed by significant propositions to continue the present strategy to fullerene generated star-branched polymers. An interesting final conclusion is that the present area of chemical science is particularly ready for digitalization. It is also notable that the concept of digital twins is about to enter modern brain research in that novel theoretical developments regarding the syntax-semantic problem in terms of algebraic, digital structures have exact polygon-polygram maps that might prove crucial for the optimal generation of modern AI-systems.
Reviewer 2 Report
Comments and Suggestions for Authors
The authors overviewed the application of spin theory for the elementary reactions of C60 fullerene which involves its free radical copolymerization with vinyl monomers and subsequently forms the backbone of the polymer process. The paper is comprehensive and well-organized, it can be taken into consideration of acceptance in the present form. General comments are provided below just for reference.
Comment 1: A reaction network can be drawn based on elementary reaction steps as listed in Table 1, so that the audience can capture a clear impression of reaction pathway.
Comment 2: A session of Perspective and Overview is recommended to be added in the main text before the Conclusion part. For example, discussions about the potential areas for future research / the bottleneck issues at this stage would provide a more balanced view.
